# DIRECT: Deep Active Learning under Imbalance and Label Noise

## Abstract

Class imbalance is a prevalent issue in real world machine learning applications, often leading to poor performance in rare and minority classes. With an abundance of wild unlabeled data, active learning is perhaps the most effective technique in solving the problem at its root – collecting a more balanced and informative set of labeled examples during annotation. Label noise is another common issue in data annotation jobs, which is especially challenging for active learning methods. In this work, we conduct the first study of active learning under both class imbalance and label noise. We propose a novel algorithm that robustly identifies the class separation threshold and annotates the most uncertain examples that are closest from it. Through a novel reduction to one-dimensional active learning, our algorithm DIRECT is able to leverage classic active learning theory and methods to address issues such as batch labeling and tolerance towards label noise. We present extensive experiments on imbalanced datasets with and without label noise. Our results demonstrate that DIRECT can save more than 60% of the annotation budget compared to state-of-art active learning algorithms and more than 80% of annotation budget compared to random sampling.

## 1 Introduction

Large-scale deep learning models are playing increasingly important roles across many industries. Human feedback and annotations have played a significant role in developing such systems. Progressively over time, we believe the role of humans in a machine learning pipeline will shift to annotating rare yet important cases. However, under data imbalance, the typical strategy of randomly choosing examples for annotation becomes especially inefficient. This is because the majority of the labeling budget would be spent on common and well-learned classes, resulting in insufficient rare class examples for training an effective model. To mitigate this issue, many recent active learning algorithms have focused on labeling more class-balanced and informative examples (Aggarwal et al., 2020; Kothawade et al., 2021; Zhang et al., 2022; 2024b; Soltani et al., 2024). For many large-scale annotation jobs, this challenge of data imbalance is further compounded by label noise – a critical and common issue that results from annotator decision fatigue and perception differences. A rich body of literature on agnostic active learning (Balcan et al., 2006; Dasgupta et al., 2007; Hanneke et al., 2014; Katz-Samuels et al., 2021) addresses this challenge on low-complexity model classes (e.g. linear models). However, for deep learning models, these algorithms often becomes ineffective due to the large model class complexity. In this paper, we propose a novel active learning strategy for both class imbalance and label noise. Our algorithm DIRECT sequentially and adaptively chooses informative and more class-balanced examples for annotation while being robust to noisy annotations. To the best of our knowledge, this is the first deep active learning study to address the challenging yet prevalent scenario where both imbalance and label noise coexist.

To bridge the gap between the imbalanced deep active learning and the agnostic active learning literature, we propose a novel reduction of the imbalanced classification problem into a set of one-dimensional agnostic active learning problems. For each class, our reduction sorts unlabeled examples into an list ordered by one-vs-rest margin scores. The objective of DIRECT is to find the *optimal separation threshold* which best separates the examples in the given class from the rest. By relating our problem to that of finding the best threshold classifier, we are able to employ ideas from the agnostic active learning literature to learn the separation threshold robustly under label noise. By annotating around the threshold, the annotated examples are more class-balanced and informative.

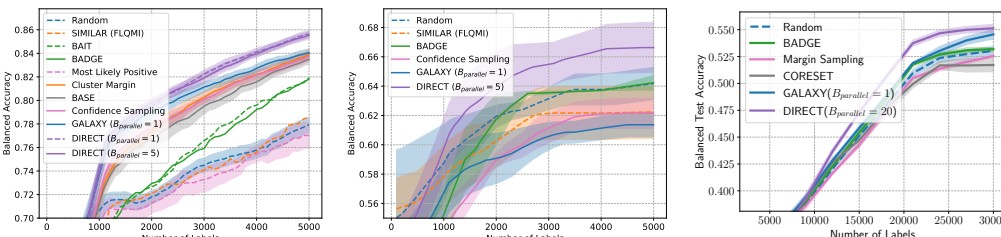

(a) Imbalanced CIFAR-10, two classes, no label noise.
(b) Imbalanced CIFAR-100, two classes, 20% label noise.
(c) LabelBench FMoW (62 classes with imbalance), no label noise

Figure 1: Performance of DIRECT over existing baselines for both noiseless and noisy settings. The x-axis represents the total number of labeled examples so far and the y-axis shows the neural network's balanced accuracy. Both (a) and (b) are using supervised training of ResNet-18. In (c), we finetune CLIP ViT-B32 model in combination of semi-supervised training under the LabelBench framework (Zhang et al., 2024a). $B_{\text{parallel}}$ is the batch size indicating the number of parallel annotators. $B_{\text{parallel}} = 1$ indicates the synchronous annotation requirement by GALAXY. Our algorithm DIRECT takes pre-specified $B_{\text{parallel}}$ as input, which is determined by real world scenarios.

Comparing to existing active learning algorithms such as BADGE (Ash et al., 2019), Cluster-Margin (Citovsky et al., 2021), SIMILAR (Kothawade et al., 2021), GALAXY (Zhang et al., 2022) and many others, DIRECT improves significantly in label efficiencies – less annotations needed to reach the same accuracy. Notably, most existing methods mentioned above are proposed to handle batch labeling, while previous work by Zhang et al. (2022) proposes a superior performance algorithm at the cost of only allowing one annotation at a time. Our algorithm DIRECT is able to obtain the best of both worlds – practical scalability to large annotation jobs by batch labeling while also getting superior performance than all algorithms including GALAXY. On imbalanced datasets, DIRECT achieves state-of-art label efficiency on both supervised fine-tuning of ResNet-18 and semi-supervised fine-tuning of large pretrained model under the LabelBench (Zhang et al., 2024a) framework.

To summarize our main contributions:

- We propose a novel reduction that bridges the advancement in the theoretical agnostic active learning literature to imbalanced active classification for deep neural networks.
- Our algorithm DIRECT addresses the prevalent imbalance and label noise issues and annotates a more class-balanced and informative set of examples.
- Compared to state-of-art algorithm GALAXY (Zhang et al., 2022), DIRECT allows parallel annotation by multiple annotators while still maintaining significant label-efficiency improvement.
- We conduct experiments across eight dataset settings, four levels of label noise and for both ResNet-18 and large pretrained model (CLIP ViT-B32). DIRECT consistently outperforms existing baseline algorithms by saving more than 60% annotation cost compared to the best existing algorithm, and more than 80% annotation cost compared to random sampling.

## 2 RELATED WORK

**Class-Balanced Deep Active Learning** Active earning strategies sequentially and adaptively choose examples for annotation. Many uncertainty-based deep active learning methods extend the traditional active learning literature such as margin, least confidence and entropy sampling (Tong & Koller, 2001; Settles, 2009; Balcan et al., 2006; Kremer et al., 2014). These methods have been shown to perform among the top when fine-tuning large pretrained models and combined with semi-supervised learning algorithms (Zhang et al., 2024a). More sophisticated methods have been proposed to optimize chosen examples' uncertainty (Gal et al., 2017; Ducoffe & Precioso, 2018; Beluch et al., 2018), diversity (Sener & Savarese, 2017; Geifman & El-Yaniv, 2017; Citovsky et al., 2021), or a mix of both (Ash et al., 2019; 2021; Wang et al., 2021; Elenter et al., 2022; Mohamadi et al., 2022). However, these methods often perform poorly under prevalent and realistic scenarios such as label noises (Khosla et al., 2022) or class imbalance Kothawade et al. (2021); Zhang et al. (2022; 2024a).

**Deep Active Learning under Imbalance** Data imbalance and rare instances are prevalent in almost all modern machine learning applications. Active learning techniques are effective in addressing the problem in its root by collecting a more class-balanced dataset (Aggarwal et al., 2020; Kothawade et al., 2021; Emam et al., 2021; Zhang et al., 2022; Coleman et al., 2022; Jin et al., 2022; Cai, 2022; Zhang et al., 2024b). To this end, Kothawade et al. (2021) propose a submodular-based method that actively annotates examples similar to known examples of rare instances. GALAXY(Zhang et al., 2022) constructs one-dimensional linear graphs and applies graph-based active learning techniques in annotating a set of examples that are both class-balanced and uncertain. While GALAXY outperforms existing algorithms, due to a bisection procedure involved, it does not allow parallel annotation. In addition, bisection procedures are generally not robust against label noises, a prevalent challenge in real world annotation tasks. Our algorithm DIRECT mitigates all of the above shortcomings of GALAXY while outperforming it even with synchronous labeling and no label noise, beating GALAXY in its own game. Lastly, we distinguish our work from Zhang et al. (2024b), where the paper studies the algorithm selection problem. Unlike our goal of proposing a new deep active learning algorithm, the paper proposes meta algorithms to choose the right active learning algorithm among a large number of candidate algorithms.

**Agnostic Active Learning for Label Noise** Label noise for active learning has been primarily studied under the extensive literature on agnostic learning. We refer the interested reader to the survey (Hanneke et al., 2014) for a thorough discussion. All of these works, beginning with the seminal works by Balcan et al. (2006); Dasgupta et al. (2007), follow a familiar paradigm of disagreement based learning. This involves maintaining a version space of promising hypotheses at each time and constructing a disagreement region of unlabeled examples. For any unlabeled example in the disagreement region, there exists two hypotheses in the version disagreeing on their predictions. An example then chosen for annotation by sampling from a informative sampling distribution computed over the disagreement region. Several approaches have been proposed for computing such sampling distributions, e.g. Jain & Jamieson (2019); Katz-Samuels et al. (2020; 2021); Huang et al. (2015). As described in Section 4.2, our main subroutine VReduce is equivalent to fixed-budget one dimensional threshold disagreement learning based on the ACED algorithm of Katz-Samuels et al. (2021). We remark that these algorithms tend to be overly pessimistic in training deep neural nets, and this paper hopes to close this gap.

**Deep Active Learning under Label Noise** Label noisy settings has rarely been studied in the deep active learning literature. Related but tangential to our work, several papers have studied to use active learning for cleaning existing noisy labels (Lin et al., 2016; Younesian et al., 2021). In this line of work, they assume access to an oracle annotator that will provide clean labels when queried upon. This is fundamentally different from our work, where our annotator may provide noisy labels. Another line of more theoretical active learning research studies active learning with multiple annotators with different qualities (Zhang & Chaudhuri, 2015; Chen et al., 2022). The primary goal in these work is to identify examples a weak annotator and a strong annotator may disagree, in order to only use the strong annotator on such instances. In our work, we assume access to a single source of annotator that is noisy, which is prevalent in annotation jobs today. Recently, Khosla et al. (2022) proposed a novel deep active learning algorithm specialized for Heteroskedastic noise, where different "regions" of examples are subject to different levels of noise. Unlike their work, our work is agnostic to the noise distributions and conduct experiments on uniformly random corrupted labels.

To our knowledge, no deep active learning literature has studied the scenario where both imbalance and label noise present. Yet, this setting is the most prevalent in real-world annotation applications.

## 3 PRELIMINARY

### 3.1 NOTATIONS

We study the pool-based active learning problem, where an initial unlabeled set of $N$ examples $X = \{x_1, ..., x_N\}$ are available for annotation. Their corresponding labels $Y = \{y_1, ..., y_N\}$ are initially unknown. Furthermore, we study the multi-class classification problem, where the space of labels $\mathcal{Y} := [K]$ is consisted of $K$ classes. Moreover, let $N_1, ..., N_K$ denote the number of examples in $X$ of each class. We define the imbalance ratio as $\gamma = \frac{\min_{k \in [K]} N_k}{\max_{k' \in [K]} N_{k'}}$.

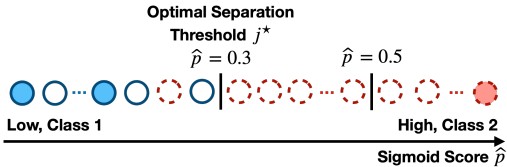
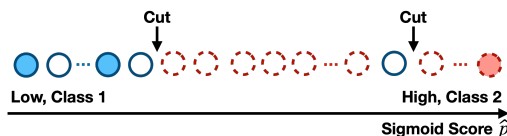

(a) Uncertainty based methods that query around $\widehat{p} =$ .5 could annotate examples only in the majority class.

(b) GALAXY spends approximately equal annotation budget around both cuts, while the cut on the right would yield examples mostly in the majority class.

Figure 2: Demonstration of existing imbalance active learning algorithms. Ordered lists of examples are ranked by the predictive sigmoid score $\widehat{p}$. The ground truth label of each example is represented by its border – solid blue for class 1 and dotted red for class 2. Annotated examples are shaded.

A deep active learning algorithm iteratively chooses batches of examples for annotation. During the $t$-th iteration, the algorithm is given labeled and unlabeled sets of examples, $L_t$ and $U_t$ respectively, where $L_t \cup U_t = X$ and $L_t \cap U_t = \emptyset$. The algorithm then chooses $B$ examples from the unlabeled set $X^{(t)} \subseteq U_t$ and then obtains their corresponding labels $Y^{(t)}$. The labeled and unlabeled sets are then updated, i.e., $L_{t+1} \leftarrow L_t \cup X^{(t)}$ and $U_{t+1} \leftarrow U_t \backslash X^{(t)}$. Based on new labeled set $L_{t+1}$ and its corresponding labels, a neural network $f_t : X \rightarrow [K]$ is trained to inform the choice for the next iteration. The ultimate goal of deep active learning is to obtain high predictive accuracy for the trained neural network while annotating as few examples as possible.

## 3.2 LIMITATIONS OF EXISTING IMBALANCED ACTIVE LEARNING ALGORITHMS

Below we document the several active learning algorithms and how their progressive improvement. At the end, we highlight the shortcomings of the state-of-art algorithm GALAXY (Zhang et al., 2022) and motivate DIRECT's objective of adaptively finding the *optimal separation threshold*. We first consider an imbalanced binary classification case, where $N_1 < N_2$ without loss of generality.

**Random Sampling.** After annotating a significant number of examples, random sampling would annotate a subset of $X$ with an imbalance ratio close to $\frac{N_1}{N_2}$. This approach suffers from annotating examples that are neither class-balanced nor informative.

**Uncertainty Sampling.** In the binary classification case, uncertainty sampling methods, such as confidence (Settles, 2009), margin (Tong & Koller, 2001; Balcan et al., 2006) and entropy (Kremer et al., 2014) sampling, simply sort examples based on their predictive sigmoid scores $\widehat{p}$ and annotate examples closest to .5 as demonstrated in Figure 2a. As shown in our results in Figure 1 and Section 5, uncertainty sampling, despite improving over random sampling, significantly underperforms DIRECT and GALAXY and consistently collects less balanced annotations. This shortcoming suggests there are significantly more majority examples than minorities around the decision boundary of $\widehat{p} = .5$.

**Objective of DIRECT.** To mitigate the above issue with the decision boundary, we propose to identify the *optimal separation threshold*. The threshold best separates the minority and majority classes and approximately equalizes the number of examples from both classes around its vicinity (see Section 4.1 for formal definition). We note the optimal separation threshold could be relatively distant from $\widehat{p} = .5$, as shown in Figure 2a. Our overall objective is to label examples that are *both uncertain and class-balanced*, and can be decomposed into the following two-phased procedure:

1. Identify the *optimal separation threshold $j^\star$* that best separates the minority class from the majority class, as shown in Figure 2a.
2. Annotate equal number of examples next to $j^\star$ from both sides.

**Limitation of GALAXY(Zhang et al., 2022).** As discussed above, the neural network decision boundary $\widehat{p} = .5$ does not necessarily best separate minority and majority class examples. GALAXY draws inspiration from graph-based active learning. It relies on the fact that the best separation threshold must be a cut, namely thresholds with a minority class example to the left and a majority class example to the right (see Figure 2b). The algorithm aims to find *all* cuts in the sorted graph as shown in Figure 2b. However, GALAXY suffers from three weaknesses:

1. During active learning, the neural network is still under training and cannot perfectly separate the two classes of examples yet. Therefore, the sorted graph could have a significant number of cuts. As an example in Figure 2b, when annotating around all of such cuts, the algorithm could waste a significant portion of the annotation budget around misclassified outliers, leading to a large number of majority class annotations.

2. Under label noise, the incorrect annotation could lead to more cuts in the sorted graph, further exacerbating the above issue.

3. GALAXY finds all cuts through a modified bisection procedure, which only allows for sequential labeling and prevents multiple annotators labeling in parallel.

In this paper, we take a DIRECT approach by identifying only the optimal separation threshold and address all of the shortcomings above.

## 4 A ROBUST ALGORITHM FOR ACTIVE LEARNING UNDER IMBALANCE AND LABEL NOISE

In this section, we formally define the optimal separation threshold and pose the problem of identifying it as an 1-dimensional reduction to the agnostic active learning problem. We then propose an algorithm inspired by the agnostic active learning literature (Balcan et al., 2006; Dasgupta et al., 2007; Hanneke et al., 2014; Katz-Samuels et al., 2021).

### 4.1 AN 1-D REDUCTION TO AGNOSTIC ACTIVE LEARNING

We start by considering the imbalanced binary classification setting mentioned in Section 3.2. When given a neural network model, we let $\widehat{p} : X \to [0, 1]$ be the predictive function mapping examples to sigmoid scores. Here, a higher sigmoid score represents a higher confidence of the example being in class 2. We sort examples by their sigmoid predictive score similar to Section 3.2. Formally, we now define the optimal separation threshold as described in Section 3.2.

**Definition 4.1.** Let $0 = q_{(0)} \leq q_{(1)} \leq \cdots \leq q_{(N)}$, where $\{q_{(i)} \in \mathbb{R}\}_{i=1}^N$ is a sorted permutation of $\{\widehat{p}(x_i)\}_{i=1}^N$. Further we let $\{x_{(i)}\}_{i=1}^N$ and $\{y_{(i)}\}_{i=1}^N$ denote the sorted list's corresponding examples and labels. We define the *optimal separation threshold* as $j^\star \in \{0, 1, ..., N\}$ such that

$$j^\star = \arg\max_j \left( |\{y_{(i)} = 1 : i \leq j\}| - |\{y_{(i)} = 2 : i \leq j\}| \right)$$

$$= \arg\max_j \left( |\{y_{(i)} = 2 : i > j\}| - |\{y_{(i)} = 1 : i > j\}| \right). \tag{1}$$

In other words, on either side of $j^*$, it has the largest discrepancy in the number of examples between the two classes. This captures the intuition of Figure 2a — our goal is to find a threshold that best separates one class from the other. We quickly remark that ties are broken by choosing the largest $j^\star$ that attains the argmax if class 1 is the minority class and the lowest $j^\star$ otherwise.

**1D Reduction.** We now provide a reduction of finding $j^\star$ to an 1-dimensional agnostic active learning problem. We define the hypothesis class $\mathcal{H} = \{h_0, h_1, ..., h_N\}$ where each hypothesis $h_j$ is defined as $h_j(q) = \begin{cases} 1 & \text{if } q \leq q_{(j)} \\ 2 & \text{if } q > q_{(j)} \end{cases}$. Here, $q_{(0)} = 0$ defines the hypothesis $h_0$ that predicts class 2 at all times. The empirical zero-one loss for each hypothesis is then defined as $\mathcal{L}(h_j) = \sum_{i=1}^N \mathbf{1}\{h_j(q_{(i)}) \neq y_{(i)}\}$. In Appendix A, we show that optimizing for the zero-one loss $\arg\min_{0 \leq j \leq N} \mathcal{L}(h_j)$ is equivalent to equation 1. Namely, with ties broken similar to above, $j^\star = \arg\min_{0 \leq j \leq N} \mathcal{L}(h_j)$.

**Multi-Class Classification.** To generalize the above problem formulation to multi-class classification, we follow a similar strategy to Zhang et al. (2022). As shown in Figure 3, for each class, we can view the

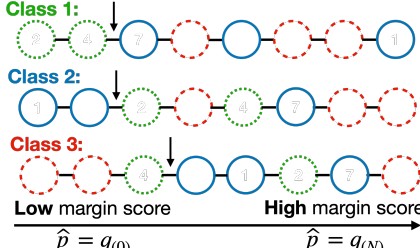

Figure 3: Visualization of multi-class classification. For each class, we formulate the problem as a one-vs-rest binary classification problem by sorting examples based on margin scores. The black arrows indicates the optimal separation thresholds for each class.

---

**Algorithm 1** DIRECT: DImension REduction for aCTive Learning under Imbalance and Label Noise

---

**Input:** Pool $X$, #Rounds $T$, retraining batch size $B_{\text{train}}$, number of parallel annotations $B_{\text{parallel}}$.

**Initialize:** Uniformly sample $B$ elements from $X$ to form $L_0$. Let $U_0 \leftarrow X \backslash L_0$.

**for** $t = 1, ..., T - 1$ **do**

    Train neural network on $L_{t-1}$ and obtain $f_{t-1}$.

    **Find optimal separation thresholds**

    Initialize labeled set $L_t \leftarrow L_{t-1}$ and budget per class $b \leftarrow B_{\text{train}}/2K$.

    **for** $k$ in RandPerm($\{1, ..., K\}$) **do**

        Sort margin scores $0 = q_{(0)}^k \leq q_{(1)}^k \leq \cdots \leq q_{(N)}^k$ based on equation equation 2.

        Let $x_{(i)}^k, y_{(i)}^k$ denote the example and label corresponding to $q_{(i)}^k$.

        Identify threshold for class $k$: $L_t \leftarrow \text{VReduce}(L_t, b, k, B_{\text{parallel}}, \{(x_{(i)}^k, y_{(i)}^k), q_{(i)}^k\}_{i=1}^N)$.

    **end for**

    **Annotate examples around the identified threshold**

    Compute budget per class $b \leftarrow (B_{\text{train}} - |L_t|)/K$.

    **for** $k$ in RandPerm($\{1, ..., K\}$) **do**

        Estimate separation threshold (break ties by choosing the index closest to $\frac{N}{2}$):

        $\widehat{j}^k \leftarrow \arg\max_j(|\{y_{(i)} = k : x_{(i)} \in L_t \text{ and } i \leq j\}| - |\{y_{(i)} \neq k : x_{(i)} \in L_t \text{ and } i \leq j\}|)$.

        Annotate $b$ unlabeled examples with sorted indices closest to $\widehat{j}^k$ and insert to $L_t$.

    **end for**

**end for**

**Return:** Train final classifier $f_T$ based on $L_T$.

---

problem of class-$k$ v.s. others as a binary classification problem. The goal therefore becomes finding all $K$ optimal separation thresholds, which is equivalent with solving $K$ 1-D agnostic active learning problems. Moreover, let $\widetilde{p} : X \to \Delta^{(K-1)}$ denote the neural network prediction function, mapping examples to softmax scores. For each class $k$, we use the margin scores $\widehat{p}_i^k := [\widetilde{p}(x_i)]_k - \max_{k'}[\widetilde{p}(x_i)]_{k'}$ to sort the examples and break ties by their corresponding confidence scores $[\widetilde{p}(x_i)]_k$. Formally,

$$\left(q_{(1)}^k \leq \cdots \leq q_{(N)}^k : \text{sorted permutation of } \{\widehat{p}_i^k\}_{i=1}^N\right) \wedge \left(q_{(i)}^k = q_{(i+1)}^k \Rightarrow [\widetilde{p}(x_i)]_k \geq [\widetilde{p}(x_{i+1})]_k\right). \quad (2)$$

Note that sorting by margin scores is equivalent to sorting by sigmoid scores for binary classification.

## 4.2 ALGORITHM

We are now ready to state our algorithm DIRECT as shown in Algorithm 1. Each round of DIRECT follows a two-phased procedure, where the first phase aims to identify the optimal separation threshold for each class. The second phase then annotates examples closest to the estimated optimal separation thresholds for each class. We spend half each round's budget for both phases.

During the first phase, to identify the optimal separation threshold for all classes, we loop over each class $k$ and run a agnostic active learning procedure for the corresponding 1-D class-k v.s. rest reduction. Our agnostic active learning subroutine is formally outlined in Algorithm 2, VReduce. Our method corresponds to the fixed-budget ACED Katz-Samuels et al. (2021) algorithm, adopted to the threshold classifier scenario. To explain the intuition about ACED in this scenario, we first note that in the separable case with $\mathcal{L}(h_{j^*}) = 0$, we could simply run a bisection procedure to learn $j_k^*$, the optimal separation threshold for the $k$-th class. However, as $j^*$ cannot perfectly separate the classes, ACED circumvents this by maintaining a version space of possible thresholds, namely the interval $[I, J]$. Statistically with high likelihood, the optimal separation threshold $j_k^*$ lies in this version space of $[I, J]$. During each of the $m$ rounds of VReduce, $B_{\text{parallel}}$ samples are annotated and the version space's length is shrank by a factor of $1/c$. The shrinkage rate $c$ is determined by the budget and batch size, so that after the final iteration, the version space has exactly one hypothesis left. The second phase of DIRECT simply annotates examples closest to each optimal separation threshold, aiming to annotate a class-balanced and uncertain examples.

To address batch labeling, we let $B_{\text{train}}$ denote the number of examples the algorithm collects before the neural network is retrained. In practice, this number is usually determined by the constraints of computational training cost. On the other hand, we let $B_{\text{parallel}}$ denote the number of examples annotated in parallel. We note that, in practice, the number of examples collected before retraining is

---

**Algorithm 2** VReduce: Version Space Reduction

---

**Input:** Labeled set $L$, budget $b$, class of interest $k$, parallel batch size $B_{\text{parallel}}$, examples and ground truth labels with sorted uncertainty $\{x_{(i)}^k, y_{(i)}^k, q_{(i)}^k\}_{i=1}^N$ (unlabeled $y_{(i)}^k$ hidden to learner).

**Initialize:** Version space as the shortest segment of indices $[I, J]$ such that: for each labeled example $x_{(i)}$ with $i \leq I$, $y_{(i)} = k$, and for each labeled example $x_{(j)}$ with $j \geq J$, $y_{(j)} \neq k$.

**Initialize:** Number of iterations $m \leftarrow \frac{b}{B_{\text{parallel}}}$. Shrinking factor $c \leftarrow \sqrt[m]{J - I}$

**for** $t = 1, ..., m$ **do**

    Uniformly sample $B_{\text{parallel}}$ unlabeled examples in $x_{(I)}^k, ..., x_{(J)}^k$ for annotation and insert to $L$.

    For each $0 \leq s \leq N$, compute $\widehat{\mathcal{L}}^k(s) = \sum_{r \leq s : x_{(r)} \in L} \mathbf{1}\{y_{(r)} \neq k\} + \sum_{r > s : x_{(r)} \in L} \mathbf{1}\{y_{(r)} = k\}$.

    Shrink version space by $\frac{1}{c}$: $I, J \leftarrow \arg\min_{i,j \in [I,J]: j-i=\frac{1}{c}(J-I)} \max\{\widehat{\mathcal{L}}^k(i), \widehat{\mathcal{L}}^k(j)\}$.

**end for**

**Return:** Updated labeled set $L$.

---

usually far greater than the number of annotators annotating in parallel, i.e., $B_{\text{parallel}} \ll B_{\text{train}}$. Lastly, as will be discussed in Section 6, our algorithm can also be modified for asynchronous labeling.

**Theoretical Comparison with GALAXY.** As mentioned in Section 3.2, GALAXY's graph-based approach aims to identify all *cuts* and sample examples around all of them equally. On the other hand, DIRECT aims to identify only the separation threshold and sample around it, which is superior as we have argued before and shown in our results. We now present a more theoretical comparison. As we show in Appendix A, the graph-based approach in GALAXY will identify and annotate around at least one more cut in addition to the optimal separation threshold, with probability at least $1 - \exp(-b\log(\frac{1}{1-\eta})/2)$. Here, $b$ is the budget of a single round of annotation and $\eta$ is the label noise ratio (see Appendix A for more details). This implies, when the budget $b$ is large, GALAXY will likely annotate around unnecessary cuts. This is in contrast with the agnostic active learning approach we take in DIRECT, where as shown by Katz-Samuels et al. (2021), the probability of misidentifying the optimal separation threshold decays exponentially w.r.t. budget $b$. In other words, with a large budget $b$, with high likelihood, DIRECT will focus its annotation around the optimal separation threshold. Lastly, time complexity analysis in Appendix C shows DIRECT's superior speed compared to BADGE and GALAXY.

## 5 EXPERIMENTS

We conduct experiments under two primary setups:

1. Supervised fine-tuning of ResNet-18 on imbalanced datasets similar to Zhang et al. (2022).
2. Fine-tuning large pretrained model (CLIP ViT-B32) with semi-supervised training strategies under the LabelBench framework (Zhang et al., 2024a).

For both evaluation setups, we first evaluate the performance of DIRECT under the noiseless setting in Section 5.1, showing its superior label-efficiency and ability to accommodate batch labeling. In Section 5.2, we evaluate deep active learning algorithms under a novel setting with both class imbalance and noisy labels. Under this setting, we also include an ablation study of the performance of DIRECT on various levels of label noises. While we highlight many results in this section, see Appendix E for complete results.

**Experiment Setups.** Our experiments utilize 10 imbalanced datasets derived from popular computer vision datasets. For the ResNet experiments, we utilize imbalanced versions of CIFAR-10, CIFAR-100 (Krizhevsky et al., 2009), SVHN (Netzer et al., 2011) and PathMNIST (Yang et al., 2021) datasets. For the LabelBench experiments, we utilize the FMoW (Christie et al., 2018) and iWildcam (Beery et al., 2021) datasets. We refer the readers to Appendix D for more details on our experiment setups.

### 5.1 EXPERIMENTS UNDER IMBALANCE, WITHOUT LABEL NOISE

For the noiseless experiment on ResNet-18, we compare against nine baselines: GALAXY (Zhang et al., 2022), SIMILAR (Kothawade et al., 2021), BADGE (Ash et al., 2019), BASE (Emam et al., 2021), BAIT (Ash et al., 2021), Cluster Margin (Citovsky et al., 2021), Confidence Sampling (Settles,

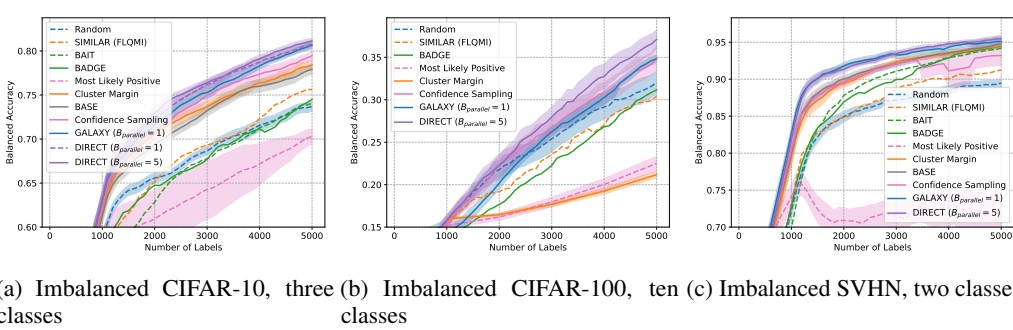

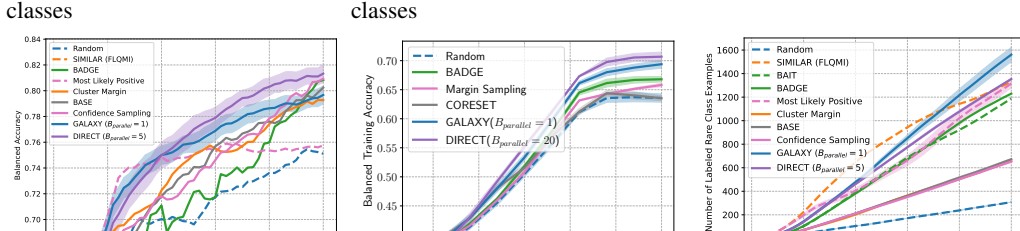

(a) Imbalanced CIFAR-10, three classes    (b) Imbalanced CIFAR-100, ten classes    (c) Imbalanced SVHN, two classes

(d) PathMNIST, two classes    (e) FMoW, Balanced Pool Accuracy    (f) Imbalanced SVHN, two classes, # of minority labeled examples

Figure 4: Performance comparison of DIRECT against other baselines algorithms in the noiseless but imbalanced setting. (a)-(d) are balanced accuracy on ResNet-18 experiments while (e) shows experiment under the LabelBench framework. $B_{\text{parallel}}$ indicates the number of parallel annotations as mentioned in Section 4.2. $B_{\text{parallel}} = 1$ is equivalent with sychornous labeling. Results are averaged over four trials and the shaded areas represent standard errors around the mean.

2009), Most Likely Positive (Jiang et al., 2018; Warmuth et al., 2001; 2003) and Random Sampling. We briefly distinguish the algorithms into two categories. In particular, both SIMILAR and Most Likely Positive annotate examples that are similar to existing labeled minority examples, thus can significantly annotate a large quantity of minority examples. The rest of the algorithms primarily optimizes for different notions of informativeness such as diversity and uncertainty. For the Label-Bench experiments, due to the large dataset and model embedding sizes, we choose a subset of the algorithms that are computationally efficient and among top performers in the ResNet-18 results, including BADGE, Margin Sampling, CORESET and GALAXY.

As highlighted in Figures 1(a) and 4(a)-(d), DIRECT consistently and significantly outperforms existing algorithms on the ResNet-18 experiments. In Figures 1(c) and 4(e), we demonstrate the increased label-efficiency is also consistently shown in the LabelBench experiments. Compared to random sampling, DIRECT can save more than 80% of the annotation cost on imbalanced SVHN experiment of Figure 4(c). In terms of class-balancedness, we consistently observe that both Most Likely Positive and SIMILAR annotating greater number of minority class examples, but significantly underperforms in terms of balanced accuracy (an example showin in Figure 4(f)). While Zhang et al. (2022) has already observed this phenomenon, we can further see that DIRECT collects slightly less minority class examples than GALAXY, but outperforms in terms of balanced accuracy. While it is crucial to optimize class-balancedness for better model performance, we see that both extremes of annotating too few and too many minority examples could lead to worse generalization performances. When too few examples are from minority class, the performance of the minority classes could be significantly hindered. When optimized to annotate as many examples from minority class as possible, the algorithm has to tradeoff annotating informative examples to examples it is more certain to be in the minority class. Together, this suggests an intricate balance between the two objectives, generalization performance and class-balancedness.

We would also like to highlight the ability to handle batch labeling. Across our experiments, we see DIRECT outperforms with different amounts of parallel annotation ($B_{\text{parallel}} = 1$, 5 and 20), indicating its general effectiveness. This is in comparison to the synchoronous nature of GALAXY, where it is always using $B_{\text{parallel}} = 1$. On Figures 1(a) and 5(a), we see that DIRECT outperforms GALAXY

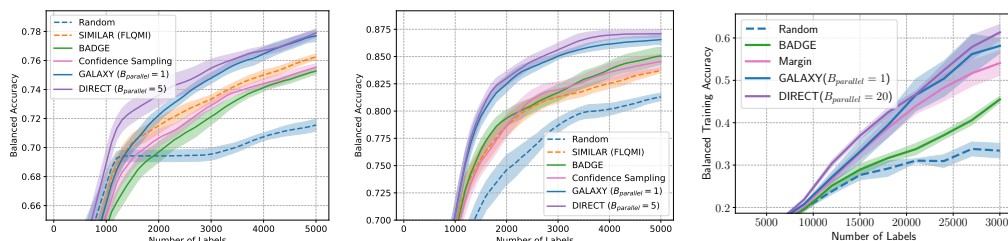

(a) Imbalanced CIFAR-10, two classes, 10% label noise

(b) Imbalanced SVHN, two classes, 10% label noise

(c) iWildcam Balanced Pool Accuracy, 10% label noise

Figure 5: Performance of DIRECT against baseline algorithms under 10% label noise. (a)-(b) are balanced accuracy on ResNet-18 experiments while (c) shows results under the LabelBench framework. Results are averaged over four trials and the shaded areas represent standard errors around the mean.

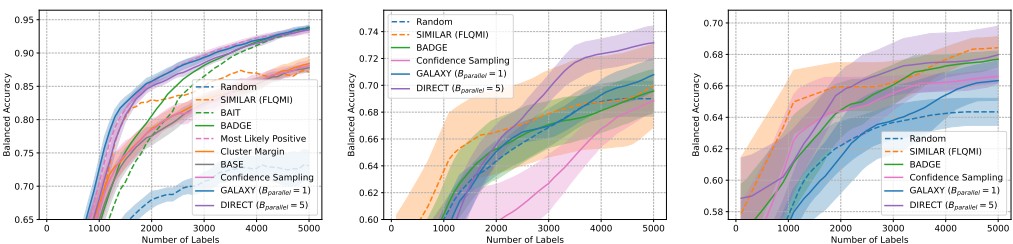

(a) Imbalanced CIFAR-100, two classes, no label noise

(b) Imbalanced CIFAR-100, two classes, 10% label noise

(c) Imbalanced CIFAR-100, two classes, 15% label noise

Figure 6: Performance of DIRECT against baseline algorithms under different levels of label noise. Results are averaged over four trials and the shaded areas represent standard errors around the mean.

with synchronous labeling. Furthermore, in these experiments we also see using $B_{\text{parallel}} = 5$ only affects algorithm performances minimally for DIRECT.

## 5.2 EXPERIMENTS UNDER IMBALANCE AND LABEL NOISE

We conduct novel sets of experiments under both class imbalance and label noise. Here, for both ResNet-18 and LabelBench experiments, we evaluate against all of the algorithms that performed well under the imbalance but noiseless setting above. For all of our experiments, we introduce a fixed percentage of label noise, where the given fraction of the examples' labels are corrupted to a different class uniformly at random. For most of our experiments with 10% label noise shown in Figure 5, we observe again that DIRECT consistently improves over all baselines including GALAXY. The results are consistent on ResNet-18 and LabelBench setups, and with different $B_{\text{parallel}}$ values, showing DIRECT's robustness under label noise.

**Different Levels of Label Noise** As shown in Figures 6(a)-(c) and 1(b), we observe the results on imbalanced CIFAR-100 with two classes across numerous levels of label noise, with 0%, 10%, 15% and 20% respectively. In fact, the noiseless experiment in Figure 6(a) is the only setting DIRECT slightly underperforms GALAXY in terms of generalization accuracy. However, we see DIRECT becomes more label-efficient under label noise. It is also worth noting that with high label noise of 20%, we observe in Figure 1(b) that existing algorithms underperform random sampling. In contrast, DIRECT significantly outperforms random sampling, saving more than 60% of the annotation cost.

## 6 CONCLUSION AND FUTURE WORK

In this paper, we conducted the first study of deep active learning under both class imbalance and label noise. We proposed an algorithm DIRECT that significantly and consistently outperforms

existing literature. In this work, we also addressed the batch sampling problem of current SOTA algorithm, GALAXY (Zhang et al., 2022), by annotating multiple examples in parallel. Studying asynchronous labeling could be a natural extension of our work. A potential solution is to utilize an asynchronous variant of one-dimensional active learning algorithm. In addition, one can further batch the labeling process across different classes to further accommodate an even larger number of parallel annotators. Lastly, we refer the readers to Appendix F for our impact statement.

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

## A    EQUIVALENT OBJECTIVE

**Lemma A.1.** *The agnostic active learning reduction is equivalently finding the optimal separation threshold. Namely,*

$$\arg\min_j \mathcal{L}(h_j) = \arg\max_j \left( |\{y_{(i)} = 1 : 1 \leq i \leq j\}| - |\{y_{(i)} = 2 : 1 \leq i \leq j\}| \right)$$

*Proof.* Recall the definitions: $h_j(q) = \begin{cases} 1 & \text{if } q \leq q_{(j)} \\ 2 & \text{if } q > q_{(j)} \end{cases}$ and $\mathcal{L}(h_j) = \sum_{i=1}^{N} \mathbf{1}\{h_j(q_{(i)}) \neq y_{(i)}\}$, we can expand the loss as follows

$$\arg\min_j \mathcal{L}(h_j) = \arg\min_j \sum_{i=1}^{N} \mathbf{1}\{h_j(q_{(i)}) \neq y_{(i)}\}$$

$$= \arg\min_j N - \sum_{i=1}^{N} \mathbf{1}\{h_j(q_{(i)}) = y_{(i)}\}$$

$$= \arg\max_j \sum_{i=1}^{N} \mathbf{1}\{h_j(q_{(i)}) = y_{(i)}\}$$

$$= \arg\max_j \left( \sum_{i=1}^{j} \mathbf{1}\{y_{(i)} = 1\} \right) + \left( \sum_{i=j+1}^{N} \mathbf{1}\{y_{(i)} = 2\} \right)$$

$$= \arg\max_j \left( \sum_{i=1}^{j} \mathbf{1}\{y_{(i)} = 1\} \right) + \left( \sum_{i=j+1}^{N} \mathbf{1}\{y_{(i)} = 2\} \right) - \left( \sum_{i=1}^{N} \mathbf{1}\{y_{(i)} = 2\} \right)$$

$$= \arg\max_j \sum_{i=1}^{j} \left( \mathbf{1}\{y_{(i)} = 1\} - \mathbf{1}\{y_{(i)} = 2\} \right)$$

$\square$

## B    THEORETICAL ANALYSIS

In this section, we analyze the performance of GALAXY under random label noise and show the probability of identifying and sampling around additional cuts increases as more examples are labeled. This is in contrast to the DIRECT's agnostic active learning approach, where the probability of identifying and sampling around only the optimal separation threshold decays exponentially in the number of labeling budget.

Specifically, under the binary classification scenario, one is given a sorted list of $N$ examples $\{x_{(i)}\}_{i=1}^{N}$, with ground truth labels $y_{(1)}^\star = y_{(2)}^\star = ... = y_{(N_1)}^\star = 1$ and $y_{(N_1+1)}^\star = ... = y_{(N_1+N_2)}^\star = 2$, where $N_1 + N_2 = N$. Under uniform i.i.d. label noise with noise ratio $\eta > 0$, the *observed labels* are denoted as $\{y_{(i)}\}_{i=1}^{N}$, where $\mathbb{P}(y_{(i)} \neq y_{(i)}^\star) = \eta$. In other words, the observed label is flipped with probability $\eta$.

**Theorem B.1.** *Given a budget of $b > 2 \log N$, let $M_b$ be the random variable denoting number of identified cuts in addition to the optimal separation threshold by one round of GALAXY. We must have $\mathbb{P}(M_b \geq 1) \geq 1 - \exp(-b \log(\frac{1}{1-\eta})/2)$, implying GALAXY samples around at least one more cut in addition to the optimal separation threshold with high probability.*

*Proof.* In the perfect scenario where GALAXY does not receive any corrupted labels, it would use $\log N$ budget with bisection to find the optimal separation threshold and annotate around it. However, within the first $\frac{b}{2}$ annotations, whenever GALAXY receives a corrupted label, it will identify a cut in addition to the optimal separation threshold, i.e., $M_b \geq 1$. Therefore, the probability of $M_b \geq 1$ is

| Name | $K$ | $N$ | Imb Ratio $\gamma = \frac{\min_k N_k}{\max_{k'} N_{k'}}$ |
|---|---|---|---|
| Imb CIFAR-10 | 2 | 50000 | .1111 |
| Imb CIFAR-10 | 3 | 50000 | .1250 |
| Imb CIFAR-100 | 2 | 50000 | .0101 |
| Imb CIFAR-100 | 3 | 50000 | .0102 |
| Imb CIFAR-100 | 10 | 50000 | .0110 |
| Imb SVHN | 2 | 73257 | .0724 |
| Imb SVHN | 3 | 54448 | .2546 |
| PathMNIST | 2 | 89996 | .1166 |
| FMoW | 62 | 76863 | .0049 |
| iWildCam | 14 | 129809 | $4.57 \cdot 10^{-5}$ |

Table 1: Dataset settings for our experiments. $N$ denotes the total number of examples in our dataset. $\gamma$ is the class imbalance ratio defined in Section 3.1.

greater than the probability of receiving at least one corrupted labels in the first $\frac{b}{2}$ annotations. With simple probability bound, we can show that

$$\mathbb{P}(M_b \geq 1) > 1 - (1 - \eta)^{b/2} = 1 - \exp(b \log(1 - \eta)/2) = 1 - \exp(-b \log(\frac{1}{1 - \eta})/2).$$

$\square$

As the theorem suggests, when $b$ is large, GALAXY will identify and annotate around at least one additional cut with high probability.

## C  TIME COMPLEXITY

The computation complexity for each batch of DIRECT is $O(KN \log(N) + B_{\text{train}} N)$ for data selection plus the training and inference costs of the neural network. $O(KN \log(N))$ comes from sorting examples by their margin scores for each class and $O(B_{\text{train}} N)$ is the cost for running Algorithm 2 for $O(B_{\text{train}})$ iterations. Each iteration of Algorithm 2 only costs $O(N)$ time as we can efficiently solve the objective by cumulative sums. We note that the cost associated with neural network training and inference is always the dominating factor.

For comparisons, BADGE has time complexity $O(B_{\text{train}} N(K + D))$, significantly more expensive than DIRECT, with $D$ denotes the dimensionality of the penultimate layer features. In addition, GALAXY has computational complexity of $O(KN \log(N)) + B_{\text{train}} KN$, also more expensive than DIRECT. In all of our experiments, both BADGE and GALAXY indeed is slower than DIRECT. We further note that the time complexity factor of $K$ in DIRECT can be easily parallelized by conducting the $K$ sorting procedures on different CPU cores.

In total, our experiments are conducted on NVIDIA 3090 ti GPUs. Each trial of the ResNet-18 experiment takes less than two hours while each trial of the LabelBench experiments takes roughly 12 hours.

## D  EXPERIMENT SETUP

**ResNet-18 Experiments.** ResNet-18 with passive training has been the standard evaluation in existing deep active literature (Ash et al., 2019; Zhang et al., 2022). Our experiment setup utilizes the CIFAR-10, CIFAR-100 (Krizhevsky et al., 2009), SVHN (Netzer et al., 2011) and PathMNIST (Yang et al., 2021) image classification datasets. The original forms of these datasets are roughly balanced across 9, 10 or 100 classes. We construct an extremely imbalanced dataset by grouping a large number of classes into one majority class. For example, given a balanced dataset above with $M$ classes. We generate an imbalanced dataset with $K$ classes ($K < M$) by the first $K - 1$ classes from the original dataset and combining the rest of the classes $K, ..., M$ into a single majority class $K$. Imbalance ratios are shown in Table 1.

For neural network training, we utilize the standard passive training on labeled examples with cross entropy loss and Adam optimizer (Kingma & Ba, 2014). The ResNet-18 model (He et al., 2016) is pretrained on ImageNet (Deng et al., 2009) from the PyTorch library. To address data imbalance, for all algorithms, we utilize a reweighted cross entropy loss by the inverse frequency of the number of labeled examples in each class. For experiments with label noise, we further add a 10% label smoothing during training (Müller et al., 2019) for all algorithms.

**LabelBench Experiments.** Proposed by Zhang et al. (2024a), LabelBench evaluates active learning performance in a more comprehensive framework. Here, we fine-tune the large pretrained model from CLIP's ViT-B32 model (Radford et al., 2021). The framework also utilizes semi-supervised learning method FlexMatch (Zhang et al., 2021) to further leverage the unlabeled examples in the pool for training. We conduct experiments on the two imbalanced datasets in LabelBench, with FMoW (Christie et al., 2018) and iWildcam (Beery et al., 2021). Similar to the ResNet-18 experiments, for all algorithms, we use a 10% label smoothing in the loss function to improve training under label noise. We did find FlexMatch to perform poorly under the combination of imbalance and label noise, so we used the passive training method for label noise experiments.

# E    ALL RESULTS

## E.1    NOISELESS RESULTS UNDER IMBALANCE

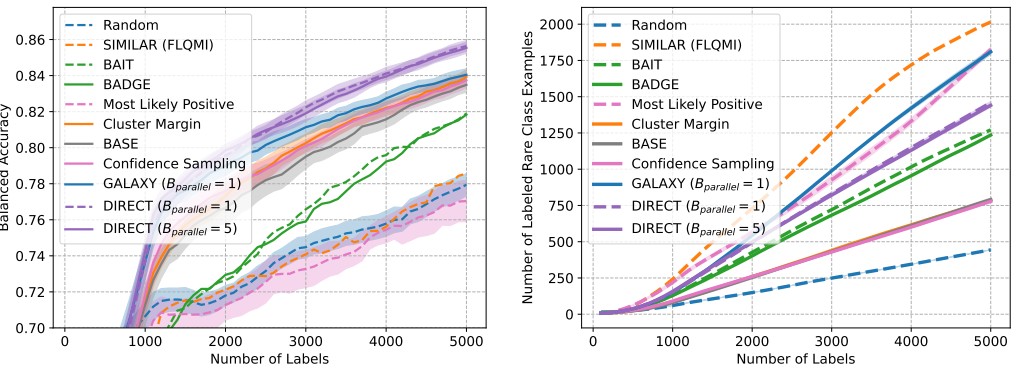

Figure 7: Imbalanced CIFAR-10, two classes.

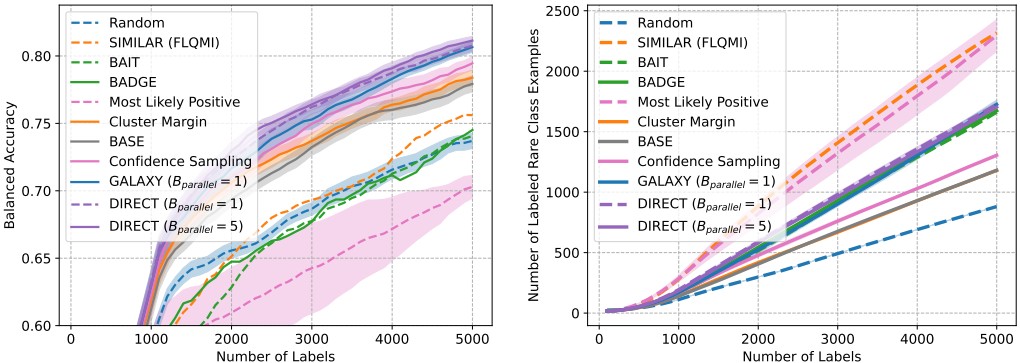

Figure 8: Imbalanced CIFAR-10, three classes.

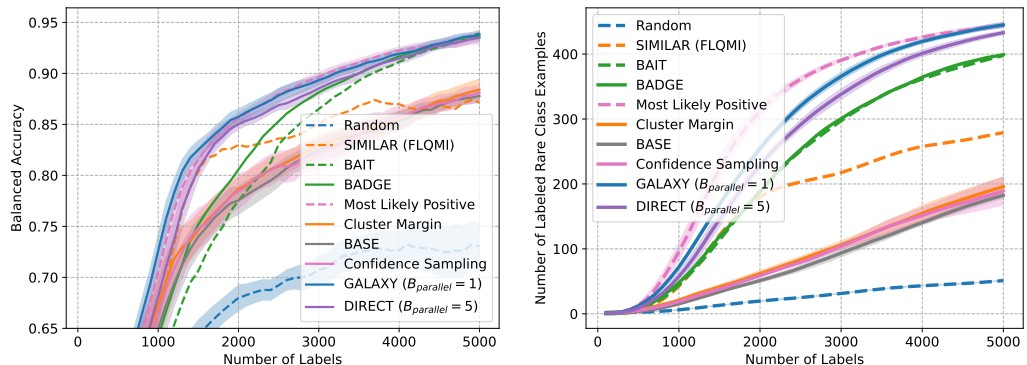

Figure 9: Imbalanced CIFAR-100, two classes.

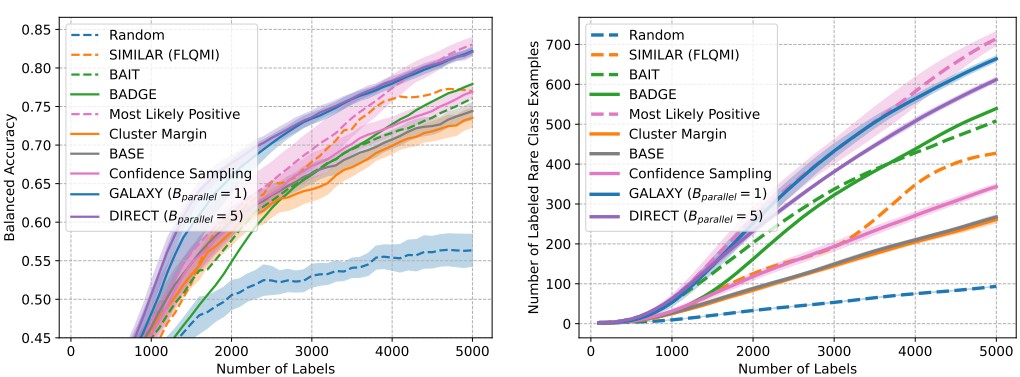

Figure 10: Imbalanced CIFAR-100, three classes.

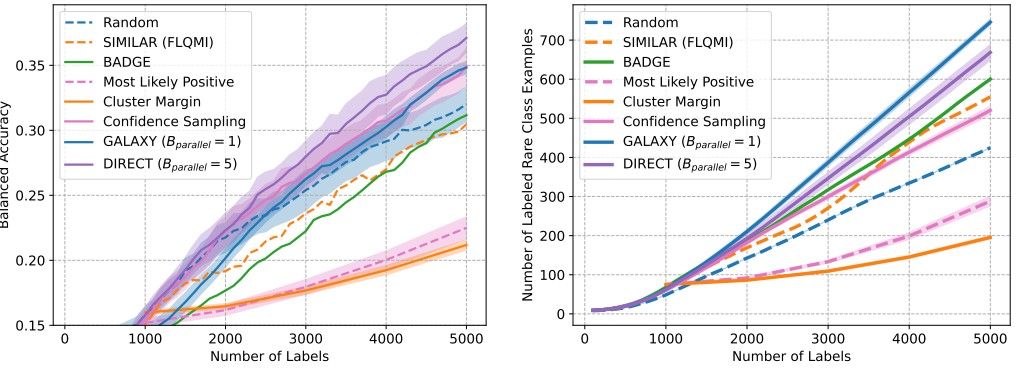

Figure 11: Imbalanced CIFAR-100, 10 classes.

### E.2 LABEL NOISE RESULTS UNDER IMBALANCE

## F IMPACT STATEMENT

In the rapidly evolving landscape of machine learning, the efficacy of active learning in addressing data imbalance and label noise is a significant stride towards more robust and equitable AI systems. This research explores how active learning can effectively mitigate the challenges posed by imbalanced datasets and erroneous labels, prevalent in real-world scenarios.

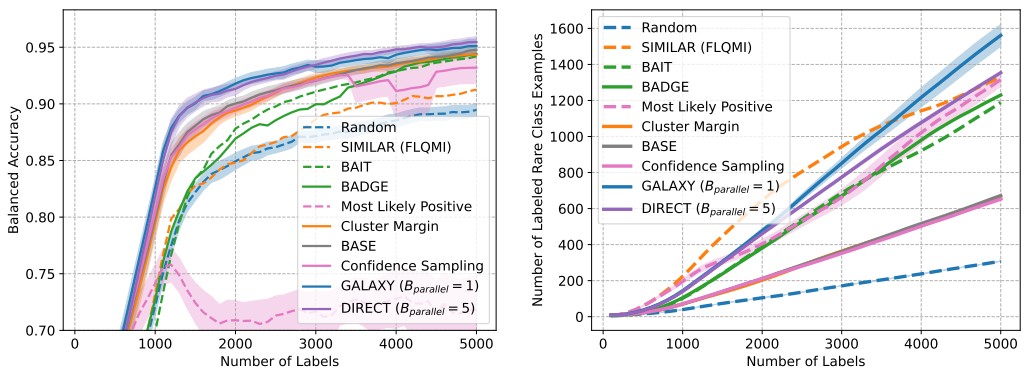

Figure 12: Imbalanced SVHN, two classes.

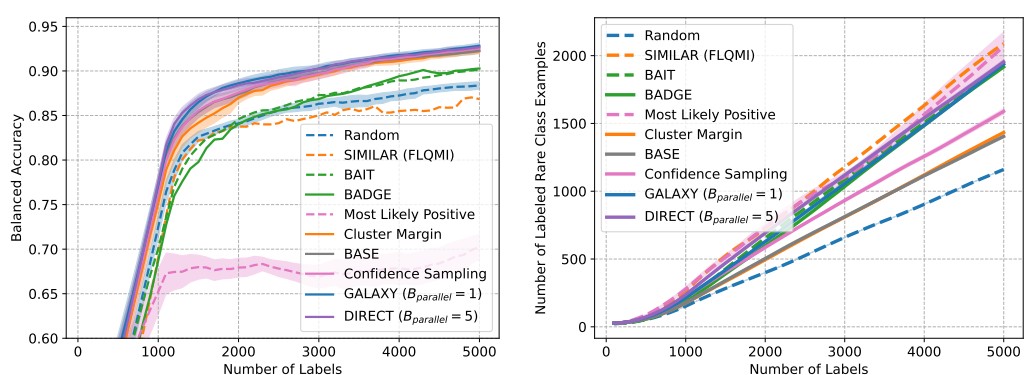

Figure 13: Imbalanced SVHN, three classes.

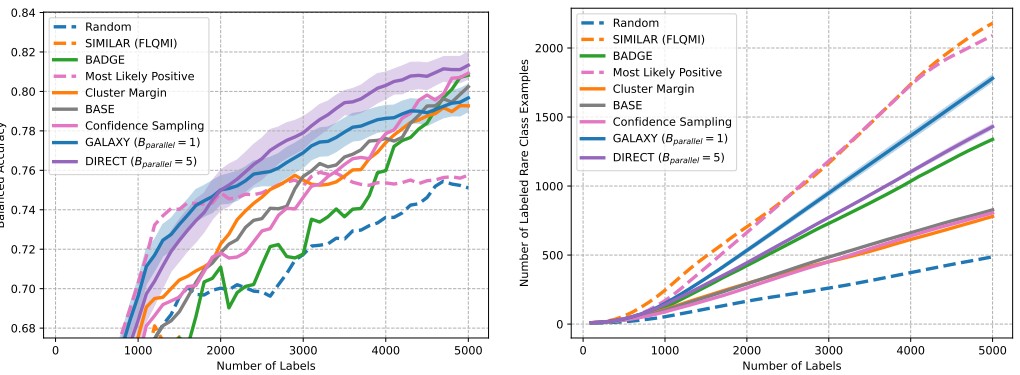

Figure 14: PathMNIST, two classes.

The positive impacts of this research are multifaceted. It enhances the accessibility and utility of machine learning in domains where data imbalance is a common challenge, such as healthcare, finance, and social media analytics. By improving class-balancedness in annotated sets, models trained on these datasets are less biased and more representative of real-world distributions, leading to fairer and more accurate outcomes. Additionally, this research contributes to reducing the time and cost associated with data annotation, which is particularly beneficial in fields where expert annotation is expensive or scarce.

However, if not carefully implemented, active learning strategies could inadvertently introduce new biases or amplify existing ones, particularly in scenarios where the initial data is severely imbalanced

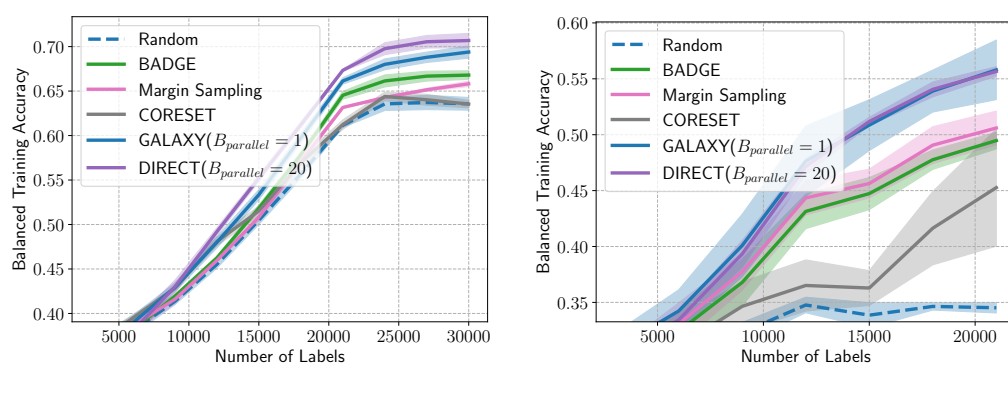

(a) FMoW Balanced Pool Accuracy

(b) iWildcam Balanced Pool Accuracy

Figure 15: LabelBench results in the noiseless setting.

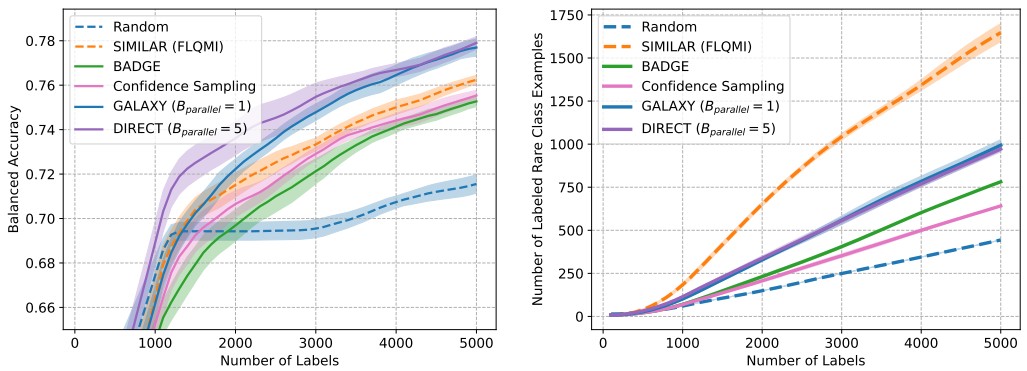

Figure 16: Imbalanced CIFAR-10, two classes, 10% label noise.

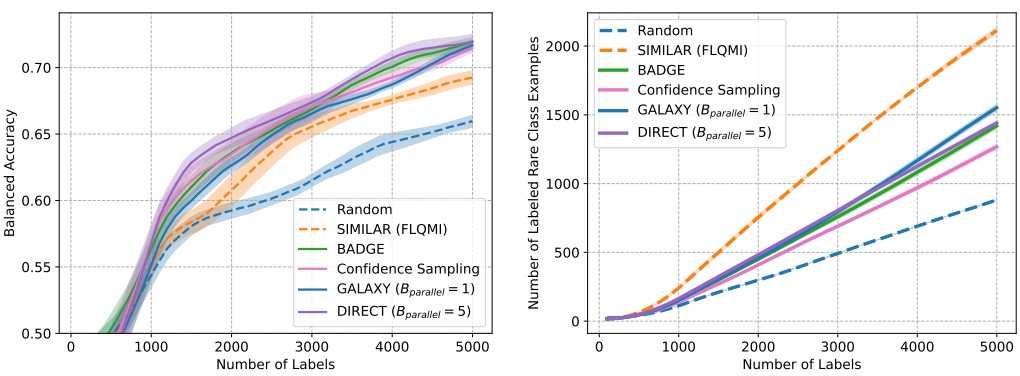

Figure 17: Imbalanced CIFAR-10, three classes, 10% label noise.

or contains deeply ingrained biases. Furthermore, the advanced nature of these techniques may widen the gap between organizations with access to state-of-the-art technology and those without, potentially exacerbating existing inequalities in technology deployment.

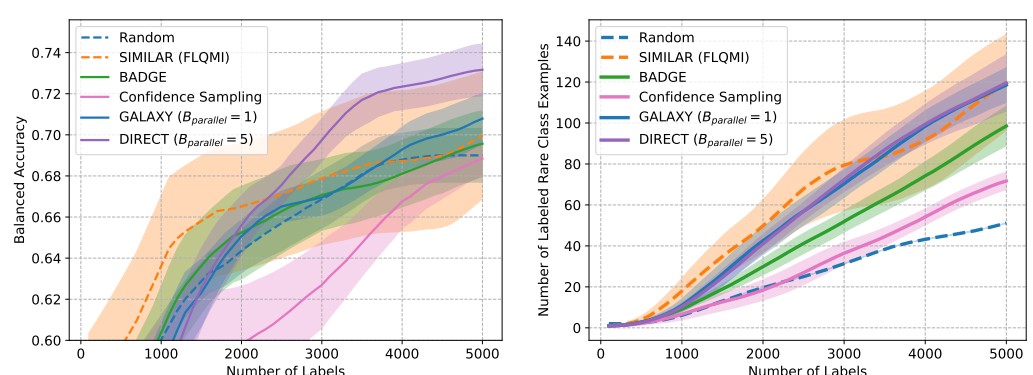

Figure 18: Imbalanced CIFAR-100, two classes, 10% label noise.

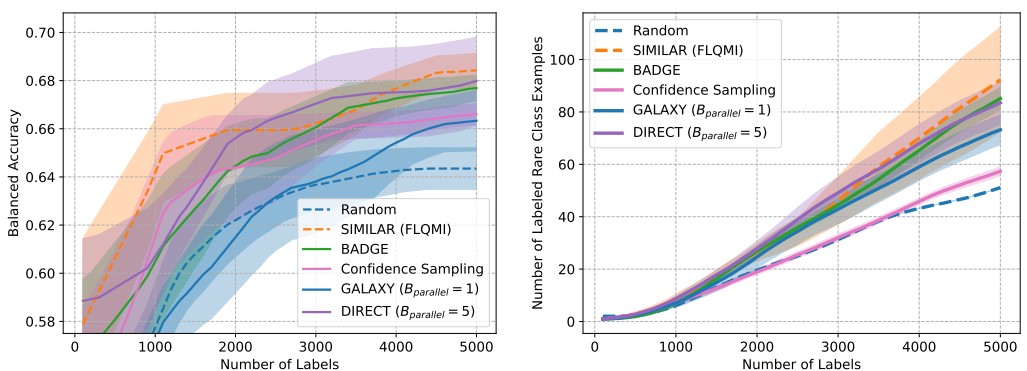

Figure 19: Imbalanced CIFAR-100, two classes, 15% label noise.

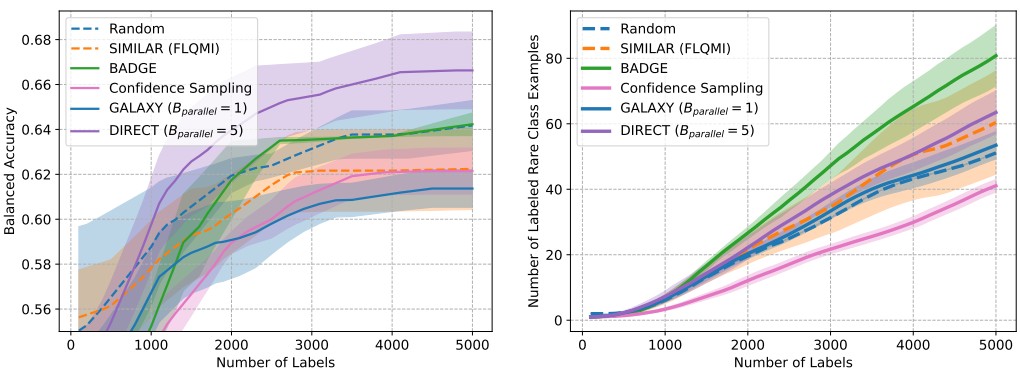

Figure 20: Imbalanced CIFAR-100, two classes, 20% label noise.

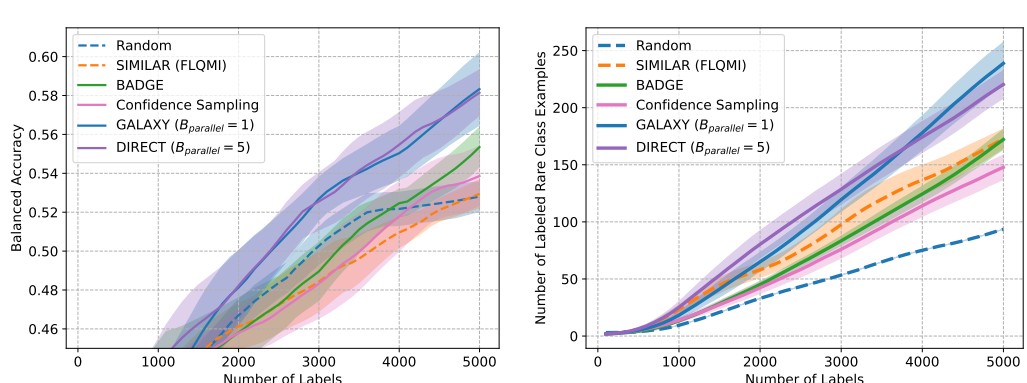

Figure 21: Imbalanced CIFAR-100, three classes, 10% label noise.

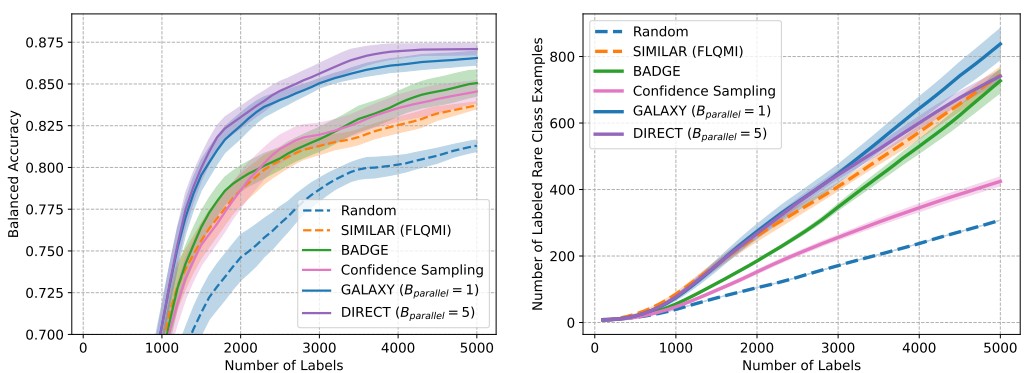

Figure 22: Imbalanced SVHN, two classes, 10% label noise.

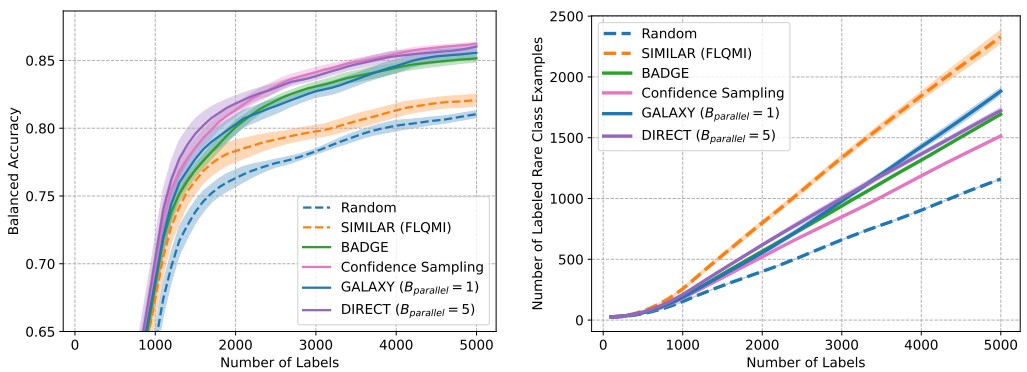

Figure 23: Imbalanced SVHN, three classes, 10% label noise.

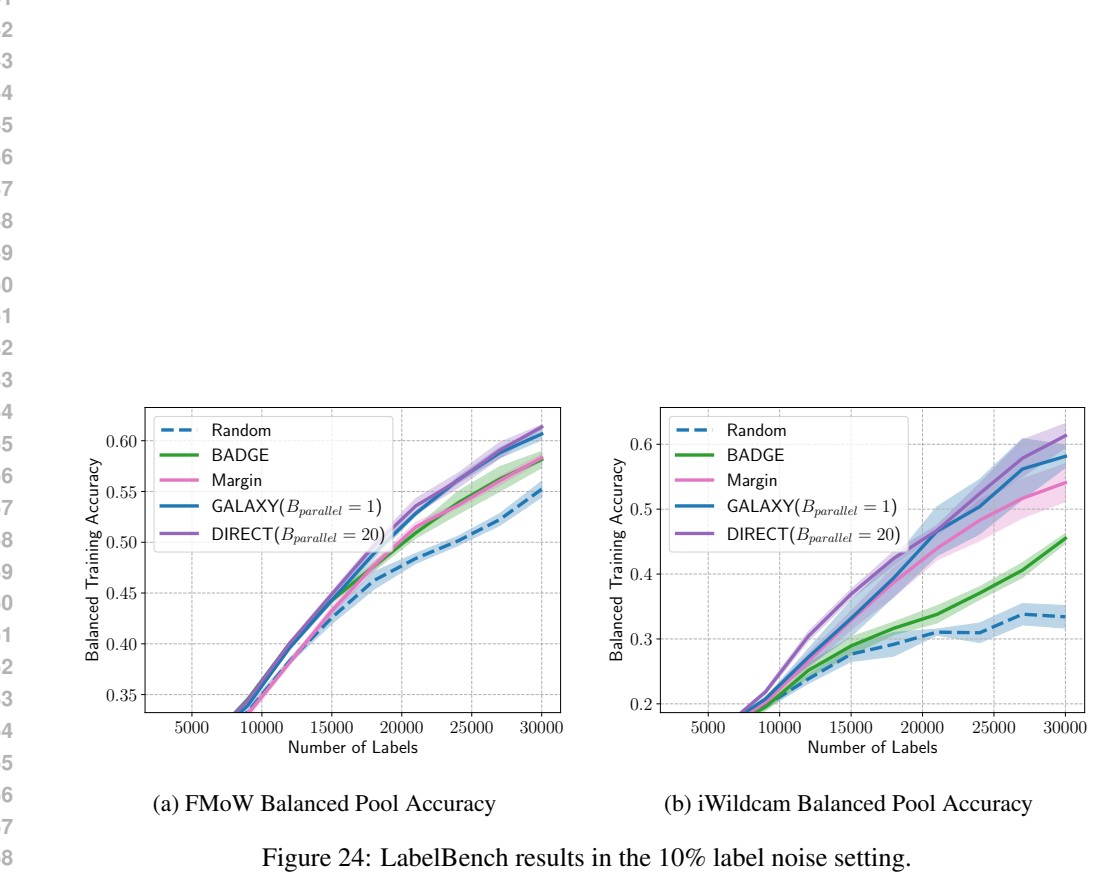

(a) FMoW Balanced Pool Accuracy

(b) iWildcam Balanced Pool Accuracy

Figure 24: LabelBench results in the 10% label noise setting.

