# OpenReview forum: "DIRECT: Deep Active Learning under Imbalance and Label Noise"
_ICLR.cc/2025/Conference — ICLR 2025 Conference Withdrawn Submission_

### Official Review · Reviewer_8brK · 2024-10-31

**Soundness:** 2
**Presentation:** 2
**Contribution:** 2
**Rating:** 3
**Confidence:** 4

**Summary:**

This paper proposes a threshold-based active learning approach aimed at handling label noise and class imbalance. The main idea for handling class imbalance is to use a deep learning method to simplify and reformulate it into a one-dimensional active learning task with a threshold learner.

**Strengths:**

1.It provides a thorough summary of some classical methods in active learning within the related work section, which is informative for readers less familiar with the field.
2.The  paper aims to address the significant problem of active learning under conditions of label noise and class imbalance which are increasingly relevant in large-scale data applications.

**Weaknesses:**

1.I believe the method proposed by the authors contains fundamental and theoretical flaws. The validity of the “sample around the optimized threshold” approach itself is highly questionable. This sampling strategy is merely an intuition; even if the authors could accurately identify this threshold, they lack any proof that sampling around it is optimal. While the authors claim this method would encourage the model to select a balanced number of samples from each class, this strategy is not necessarily the best. From a learning theory perspective, as with all active learning methods, this sampling process distorts the training distribution, potentially undermining the learning guarantees. Furthermore, from a statistical learning perspective, if the majority class has a more complex distribution, naturally more data would be required to approximate the distribution.
2.The so-called "optimal threshold" is based on true labels, which are inherently inaccessible in active learning, making accurate computation of this threshold infeasible in practice. Moreover, their hypothesis class is defined solely as a threshold classifier over the output of a deep learner. Without analyzing the behavior of this deep learning output and the underlying data distribution, any claims regarding the behavior of a hypothesis learned by empirical risk minimization (ERM) within this hypothesis class are, in my view, meaningless. Therefore, their proof is fundamentally flawed and fails to establish any meaningful guarantee for the proposed threshold and sampling strategy in active learning contexts. In fact, I think the authors’ proof in the appendix only establishes the equivalence between the ERM solution on the training data based on the learner’s output and an "optimal threshold" on this same training set. This is unrelated to what they actually need to prove—that this solution can serve as an optimal threshold for active learning on the unlabeled data.
3. The authors claim to address label noise in addition to class imbalance; however, the paper lacks even a formal definition of label noise, and it remains unclear how label noise is actually addressed. Despite incorporating noisy scenarios in their experiments, the authors do not propose any specific strategies to mitigate or manage noisy labels, leaving it ambiguous how their method effectively handles this issue.

**Questions:**

The proposed approach relies on an "optimal threshold" derived from the training set, but this threshold is based on true labels, which are inaccessible during active learning. Given that this threshold may not generalize to the unlabeled dataset, how can we justify that the threshold remains optimal or even effective for active learning purposes, especially without analyzing data distribution shifts between labeled and unlabeled sets?

---

### Official Review · Reviewer_vvC3 · 2024-11-01

**Soundness:** 2
**Presentation:** 2
**Contribution:** 2
**Rating:** 6
**Confidence:** 3

**Summary:**

This paper presents a novel active learning strategy called DIRECT, which performs well under conditions of class imbalance and label noise. Specifically, the algorithm effectively identifies the optimal class separation threshold and adaptively selects samples for annotation. Experimental results demonstrate that DIRECT significantly improves labeling efficiency, providing an effective solution for scenarios affected by class imbalance and label noise.

**Strengths:**

1. The approach of combining class separation thresholds with one-dimensional active learning is innovative and provides a fresh perspective on tackling these issues.

2. The extensive experiments conducted on imbalanced datasets provide a strong foundation for the claims made. The results indicating a 60% to 80% reduction in annotation budgets are compelling and demonstrate the algorithm's effectiveness compared to existing methods.

3. The paper is well-organized, with clear explanations of the methodology and results, making it accessible to readers.

**Weaknesses:**

1. There is a formatting error at the bottom of page 5 that requires correction to improve the document’s presentation.

2. The authors are encouraged to include additional visualizations to more effectively demonstrate the experimental results and clarify the method’s performances.

3. The authors identified three limitations in the GALAXY method and addressed these by optimizing the separation threshold. It would be beneficial to provide a more detailed explanation of this approach to help readers better understand the proposed improvements.

4. The authors demonstrated the robustness of their method under class imbalance and label noise conditions. Please clarify the sources of robustness in each scenario to highlight the model’s adaptability.

5. In the experimental section, the authors only compare label noise levels at 0%, 10%, 15%, and 20%, leaving out 5%. It would strengthen the study’s comprehensiveness to either include results at 5% noise or explore higher noise levels to make the noise experiments more comparable and thorough.

**Questions:**

See the above Weaknesses.

---

### Official Review · Reviewer_hzNB · 2024-11-03

**Soundness:** 3
**Presentation:** 2
**Contribution:** 1
**Rating:** 3
**Confidence:** 4

**Summary:**

This paper introduces an active learning algorithm designed to handle both class-imbalance and label noise in classification tasks. The proposed method reformulates the multi-class classification problem into multiple one-vs-rest tasks and employs the VReduce algorithm to estimate classification thresholds for each class. Data points near these thresholds are then selected for querying. Experiments are conducted to demonstrate the effectiveness of the approach.

**Strengths:**

The study tackles a complex and meaningful setting where both class imbalance and label noise are present, which is a valuable and practical area of focus for active learning. The idea of identifying and querying near the decision boundary, particularly for minority classes, is technically sound and shows promise for improving classification performance in imbalanced data contexts. The experiment includes recent active learning methods in its comparative analysis.

**Weaknesses:**

1)	My primary concern is the limited technical contribution. The proposed method largely leverages previously established algorithms, and the theoretical contributions appear modest.

2)	Although the paper claims to address label noise, the algorithm itself does not explicitly manage or mitigate label noise. I think the noise may relate to the agnostic learning, but in section 5.2 in the experiment, the authors conduct experiments with different levels of label noise, which confuses me.

3)	Estimating the prediction threshold for minority classes could be challenging due to the limited sample sizes within these classes. This issue is not thoroughly addressed in the paper.

4)	The paper would benefit from further proofreading to address minor errors and improve readability. For instance, lines L269 and L278 contain typographical or presentation mistakes that should be corrected.

**Questions:**

1)	About the construction of the imbalanced datasets, I am unclear about the rationale behind grouping certain classes into a larger class to create an imbalanced dataset, as this results in an inconsistent number of classes compared to the original dataset. A simpler approach, such as using a pre-existing imbalanced dataset like CIFAR-10-LT or selectively sampling data to form a minority class, might have been more straightforward. Could the authors explain the motivation behind this grouping strategy?

2)	The methods used for comparison appear inconsistent across different datasets. Could the authors clarify the criteria for choosing comparison methods, and explain any reasons for these discrepancies?

3)	When plotting the learning curves, is the budget spent on B_{parallel} included in the overall labeling budget? Clarification on this point would help in interpreting the learning curves accurately?

---

### Official Review · Reviewer_eCU7 · 2024-11-04

**Soundness:** 2
**Presentation:** 2
**Contribution:** 2
**Rating:** 3
**Confidence:** 4

**Summary:**

The paper proposes an AL strategy that deals with label imbalance and noises. The proposed method uses separation thresholds similar to an existing method GALAXY, however could enable parallel annotations and be robust against label noises.  The proposed method is compared with GALAXY and other AL baselines.

**Strengths:**

1. The paper addresses an important problem of label imbalance and noises in AL.
2. The paper improves upon existing methods like GALAXY.

**Weaknesses:**

1. The presentation of the paper is poor. While there is additional content in the appendix, the paper does not utilize the main page limit well. The figure caption of Figure 3 overlaps with main text. The algorithm is not clearly presented with too much text and no numbers for equations.
2. The results are presented poorly and unreliable. It is unclear where the AL starts and the different starting points of curves are confusing. It is also difficult to see the difference between variants of the proposed method and GALAXY in some cases.

**Questions:**

What is the starting point for AL and how to explain the initial differences?

---

### Note · Authors · 2024-11-22

I have read and agree with the venue's withdrawal policy on behalf of myself and my co-authors.